# Proscillaridin A Sensitizes Human Colon Cancer Cells to TRAIL-Induced Cell Death

**DOI:** 10.3390/ijms23136973

**Published:** 2022-06-23

**Authors:** Manami Semba, Shinji Takamatsu, Sachiko Komazawa-Sakon, Eiji Miyoshi, Chiharu Nishiyama, Hiroyasu Nakano, Kenta Moriwaki

**Affiliations:** 1Department of Biochemistry, Graduate School of Medicine, Toho University, Ota-ku, Tokyo 143-8540, Japan; 8321530@ed.tus.ac.jp (M.S.); sachiko.komazawa@med.toho-u.ac.jp (S.K.-S.); hiroyasu.nakano@med.toho-u.ac.jp (H.N.); 2Department of Biological Science and Technology, Faculty of Advanced Engineering, Tokyo University of Science, Katsushika-ku, Tokyo 125-8585, Japan; chinishi@rs.tus.ac.jp; 3Department of Molecular Biochemistry and Clinical Investigation, Graduate School of Medicine, Faculty of Medicine, Osaka University, Suita 565-0871, Osaka, Japan; shinjit@sahs.med.osaka-u.ac.jp (S.T.); emiyoshi@sahs.med.osaka-u.ac.jp (E.M.)

**Keywords:** TRAIL, apoptosis, cardiac glycoside, *O*-glycosylation

## Abstract

Tumor necrosis factor-related apoptosis-inducing ligand (TRAIL) is a cytotoxic cytokine that induces cancer cell death by binding to TRAIL receptors. Because of its selective cytotoxicity toward cancer cells, TRAIL therapeutics, such as recombinant TRAIL and agonistic antibodies targeting TRAIL receptors, have garnered attention as promising cancer treatment agents. However, many cancer cells acquire resistance to TRAIL-induced cell death. To overcome this issue, we searched for agents to sensitize cancer cells to TRAIL-induced cell death by screening a small-molecule chemical library consisting of diverse compounds. We identified a cardiac glycoside, proscillaridin A, as the most effective TRAIL sensitizer in colon cancer cells. Proscillaridin A synergistically enhanced TRAIL-induced cell death in TRAIL-sensitive and -resistant colon cancer cells. Additionally, proscillaridin A enhanced cell death in cells treated with TRAIL and TRAIL sensitizer, the second mitochondria-derived activator of caspase mimetic. Proscillaridin A upregulated TRAIL receptor expression, while downregulating the levels of the anti-cell death molecules, cellular FADD-like IL-1β converting enzyme-like inhibitor protein and Mcl1, in a cell type-dependent manner. Furthermore, proscillaridin A enhanced TRAIL-induced cell death partly via *O*-glycosylation. Taken together, our findings suggest that proscillaridin A is a promising agent that enhances the anti-cancer efficacy of TRAIL therapeutics.

## 1. Introduction

Tumor necrosis factor (TNF)-related apoptosis-inducing ligand (TRAIL) is a death ligand that belongs to the TNF superfamily. It is expressed as a type II transmembrane protein in various cell types, including cytotoxic T and natural killer cells [1]. TRAIL on these cytotoxic cells plays an important role in tumor immune surveillance by inducing tumor cell death by binding to the TRAIL receptors, death receptors 4 (DR4) and 5 (*DR5*). Recombinant soluble TRAIL successfully eliminates tumors with negligible toxicity to normal tissues in tumor xenograft mouse models [2,3]. This selective tumoricidal activity has led to the development of various types of TRAIL receptor agonists, some of which have been tested for their anti-tumor efficacy in clinical trials. In addition, recent unbiased clustered regularly interspaced palindromic repeat (CRISPR)-based genome-wide loss-of-function screens have identified the TRAIL-TRAIL receptor axis as a critical determinant of the therapeutic response of cancer cells to CAR-T cell immunotherapy [4,5]. As such, the TRAIL-TRAIL receptor axis has received considerable attention in cancer therapy. However, pre-clinical and clinical studies have revealed that many cancer cells are resistant to TRAIL receptor-mediated cell death and that this resistance is mediated by multiple mechanisms.

In response to TRAIL stimulation, TRAIL receptors form higher-order receptor clusters that recruit the adaptor protein, Fas-associated via death domain (FADD), and caspase 8 [6]. In this receptor complex, referred to as the death-inducing signaling complex (DISC), caspase 8 is activated via close proximity-induced auto-cleavage [7]. Activated caspase 8 then cleaves caspase 3 directly or via the mitochondrial pathway, which is tightly regulated by the B-cell lymphoma-2 (Bcl-2) family members, such as pro-apoptotic Bcl-2-associated X (Bax) and Bcl-2 antagonist/killer (Bak), and anti-apoptotic Mcl1 and Bcl-extra-large (Bcl-xL). Caspase-mediated signaling is regulated by various signaling molecules and post-translational modifications. For instance, ubiquitination by DISC-associated E3 ligases, such as cellular inhibitor of apoptosis 1 (cIAP1), linear ubiquitin chain assembly complex, and TNF receptor-associated factor 2 (TRAF2), inhibits caspase 8 activation, which is counteracted by the deubiquitinating enzyme, CYLD [8,9,10,11]. The expression of TRAIL receptors and these signaling proteins varies in cancer cells, which partly contributes to TRAIL resistance. 

Glycosylation is a modification of proteins and lipids that is involved in many cellular events and thereby affects tumor development and metastasis. Glycans affect signaling via death receptors, including TRAIL receptors [12]. Polypeptide *N*-acetylgalactosaminyltransferase 14 (*GALNT14*) and core 1 synthase glycoprotein-*N*-acetylgalactosamine 3-beta-galactosyltransferase 1 (*C1GALT1*)-specific chaperone 1 (*C1GALT1C1*), which are involved in protein *O*-glycosylation, and promote TRAIL-induced caspase 8 activation and apoptosis via enhanced receptor clustering and DISC formation [13,14,15]. The expression and structure of glycans significantly changes during tumor progression and are highly variable among tumor types [16], implying that glycans are partly responsible for the differential TRAIL sensitivity of various cancer cells. 

The identification of secondary agents that can overcome TRAIL resistance and enhance TRAIL-induced cancer cell death has been a long-standing interest in cancer research. In this study, we screened thousands of chemical compounds and found that proscillaridin A significantly enhanced TRAIL-induced cell death. Mechanistically, it upregulated the expression of TRAIL receptors and reduced the expression levels of anti-apoptotic proteins in a cell-type-specific manner. In addition, it increased the expression levels of enzymes involved in the synthesis of *O*-glycans, which partly contributed to the TRAIL-sensitizing effect. These results suggest that proscillaridin A is a promising agent that enhances the anticancer therapeutic efficacy of TRAIL receptor agonists and can potentially be used in immune therapy.

## 2. Results

### 2.1. Drug Screening Identified Several Agents Sensitizing Cells to TRAIL-induced Apoptosis

To select the appropriate cells for screening, we examined seven colon cancer cell lines (HT29, SW480, LS174T, Lovo, SW620, HCA7, and Caco2) for their sensitivity to TRAIL-induced cell death. Four of them, HT29, SW480, LS174T, and, to a lesser extent, Lovo, were sensitive to TRAIL-induced cell death, as the number of annexin V-positive cells was increased after treatment with 25 ng/mL of TRAIL (Figure 1A; Appendix A). In contrast, SW620, HCA7, and Caco2 cells were highly resistant to TRAIL-induced cell death (Figure 1A,B; Appendix A). We selected HT29 cells for screening as they showed the potential to die at low TRAIL concentrations when combined with secondary agents. Second mitochondria-derived activator of caspase (SMAC) mimetics are a class of cell death-inducing agents that target the inhibitor of apoptosis (IAP) family members, such as cIAP1, cIAP2, and X-linked inhibitor of apoptosis (XIAP), for proteasomal degradation, and have been evaluated for their anti-cancer efficacy in clinical trials as single agents or in combination with various therapeutics, including TRAIL therapeutics [17]. Indeed, the SMAC mimetic, birinapant, substantially sensitized HT29 cells to TRAIL-induced cell death (Figure 1C). Therefore, we aimed to identify agents that sensitized HT29 cells to TRAIL-induced cell death, even in the presence of SMAC mimetics. 

To identify these agents, we screened a small-molecule chemical library consisting of 4904 diverse compounds. HT29 cells were treated with 5 µM of each compound for 24 h and subsequently subjected to co-treatment with 1.5 ng/mL TRAIL and 0.5 µM birinapant, which caused approximately 20% reduction in cell viability without any additional compounds (Figure 1C). The cell viability was determined by using an ATP-based cell viability assay. The top 40 compounds that met the following criteria are shown in Figure 1D: (i) viability of the cells treated with a compound alone was more than approximately 70% of the viability of untreated cells and (ii) viability of the cells treated with a compound in combination with TRAIL and birinapant was less than approximately 50% of that treated with a compound alone. Three of the hit compounds, cycloheximide, dactinomycin, and emetine dihydrochloride, were transcriptional or translational inhibitors (Figure 1D). Inhibition of de novo protein synthesis by these compounds enhances TRAIL-induced cell death via downregulation of expression levels of anti-apoptotic proteins with short half-life, such as cFLIP and Mcl1 [18,19]. These results highlight the validity of our screening results. 

### 2.2. Proscillaridin A Potently Enhances TRAIL-Induced Cell Death in Colon Cancer Cells

Next, we validated the cell death-promoting effect of the 21 hit compounds, which were selected based on the literature review and commercial availability. Instead of dihydrocelastrol and dihydrocelastryl diacetate, celastrol was used for validation as both were derivatives of celastrol. HT29 cells were treated with 1 µM of each compound for 24 h and then stimulated with TRAIL. In this validation assay, the concentration of each compound was reduced to 1 µM in an attempt to identify compounds with a potent TRAIL-sensitizing activity. We stimulated the cells without SMAC mimetic to first identify compounds enhancing TRAIL sensitivity. To directly evaluate cell death rather than measuring intracellular ATP levels, we monitored the uptake of a cell-impermeable DNA-binding dye, propidium iodide (PI), during the course of the stimulation. Fifteen of the 21 compounds did not enhance TRAIL-induced cell death at 1 µM (Figure 2A) or even at higher concentrations (Appendix A). Six out of the 21 compounds (CP-100356, mitoxantrone, BIO-acetoxime, adapalene, celastrol, and proscillaridin A) caused negligible toxicity when used alone, but enhanced TRAIL-induced cell death (Figure 2A). Among the six compounds, proscillaridin A remarkably enhanced TRAIL-induced cell death. As proscillaridin A had the strongest cell death-promoting effect, we focused on characterizing its TRAIL-sensitizing effect. We treated the cells with another bivalent SMAC mimetic, BV6, to ensure that proscillaridin A enhanced TRAIL-induced cell death in the presence of a different type of SMAC mimetic. We found that proscillaridin A significantly enhanced TRAIL-induced cell death in both the absence and presence of either BV6 or birinapant (Figure 2B; Appendix A). A dose titration experiment revealed that proscillaridin A significantly enhanced TRAIL-induced cell death at concentrations as low as 11.1 nM in HT29 cells (Figure 2C; Appendix A). In addition, proscillaridin A significantly enhanced TRAIL-induced cell death in the TRAIL-sensitive colon cancer cell line, SW480, and TRAIL-resistant cell line, SW620 (Figure 2D; Appendix A). In SW620 cells, proscillaridin A significantly enhanced TRAIL-induced cell death at concentrations as low as 3.7 nM (Figure 2C; Appendix A). Moreover, a low dose of proscillaridin A (11.1 nM) enhanced TRAIL-induced cell death in the presence of the SMAC mimetic in HT29 and SW620 cells (Figure 2E; Appendix A). Whereas TRAIL-induced cell death was significantly enhanced when HT29 cells were pretreated with proscillaridin A for 24 h, such a TRAIL-sensitizing effect was not observed when pretreated for 1 h (Appendix A). These results indicate that proscillaridin A acts a potent sensitizer of TRAIL-induced cell death in the absence and presence of SMAC mimetics. 

### 2.3. Proscillaridin A Upregulates the Cell Surface Expression of TRAIL Receptors in a Cell Type-specific Manner

To delineate the underlying mechanism of proscillaridin A-mediated sensitization to TRAIL-induced cell death, we first examined the cell surface expression of the TRAIL receptors, DR4 and *DR5*. Proscillaridin A upregulated the cell surface expression of DR4 in SW480 cells, but not HT29 and SW620 cells (Figure 3A). It also upregulated the cell surface expression of *DR5* in HT29 and SW480 cells. In contrast, proscillaridin A treatment caused only a minor increase in *DR5* levels in SW620 cells, although it was statistically significant (Figure 3A). As proscillaridin A most potently enhanced TRAIL-induced cell death in SW620 cells among the three cell lines, its TRAIL-sensitizing effects on SW620 cells is likely not attributed to receptor expression. Those increases of DR4 and *DR5* expression at the cell surface were accompanied by the transcriptional upregulation, except for *DR5* in SW480 cells (Figure 3B). We also examined whether other hit compounds, adapalene, BIO-acetoxime, celastrol, and mitoxantrone, upregulated the expression levels of DR4 and *DR5* and found that none of them upregulated the expression levels of TRAIL receptors (Appendix A). These results indicate that proscillaridin A upregulates the cell surface expression of DR4 and *DR5* in a cell type-specific manner. 

### 2.4. Proscillaridin A Alters the Expression of Apoptosis-Related Signaling Proteins in a Cell Type-Specific Manner

Caspase 8 is an initiator caspase that is activated following the ligation of TRAIL-to-TRAIL receptors. In HT29 and SW620 cells, pro-caspase 8 expression levels were unchanged after proscillaridin A treatment (Figure 4A,B, lanes 1 vs. 6). In HT29 cells, caspase 8 activation, which is marked by the appearance of intermediate processed forms, was increased when stimulated with TRAIL and proscillaridin A compared to TRAIL alone (Figure 4A, lanes 4 and 5 vs. 9 and 10). In contrast, proscillaridin A did not enhance the TRAIL-induced processing of caspase 8 in SW620 cells (Figure 4B, lanes 4 and 5 vs. 9 and 10). Activated caspase 8 cleaves caspase 3 directly or via the mitochondrial pathway, which leads to the appearance of the p19 fragment. This truncated fragment is further converted to p17, the fully mature form, via autocatalytic processing [20]. Combination treatment with TRAIL and proscillaridin A produced substantially more caspase 3 p17 than treatment with TRAIL alone in HT29 and SW620 cells (Figure 4A,B). In addition, the levels of cleaved poly (ADP-ribose) polymerase (PARP), which is a marker of caspase 3 activation, were increased when the cells were treated with proscillaridin A and TRAIL compared to TRAIL alone. (Figure 4A,B).

Caspase 8 is activated in membranous and cytosolic complexes, called DISC and complex II, respectively, and is regulated by various signaling proteins recruited to these complexes [11]. We examined the expression of these regulatory proteins following treatment with proscillaridin A. The expression of FADD, cIAP1, RIPK1, TRAF2, IKKα/β, and CYLD was not affected by proscillaridin A in any of the tested cell lines (Figure 4A–C). We found that the protein expression of cellular FADD-like IL-1β converting enzyme-like inhibitor protein long isoform (cFLIP_L_) was reduced in SW480 cells (Figure 4C), whereas its mRNA expression was not reduced (Figure 4D). As cFLIP_L_ is an enzymatically inactive homolog of caspase 8 that inhibits caspase 8 activation [21], this reduction in cFLIP_L_ protein expression may contribute to TRAIL sensitization by proscillaridin A in SW480 cells. The expression of XIAP, which binds to and inhibits caspase 3, did not change after proscillaridin A treatment. We further examined the expression of proteins related to the mitochondrial pathway, including Mcl1, Bcl-xL, and Bax (Figure 4C). Mcl1 and Bcl-xL are negative regulators of the pro-apoptotic Bcl family members, Bax and Bak [22]. Proscillaridin A treatment caused a reduction in the protein expression levels of Mcl1, but not Bcl-xL, in SW620 cells (Figure 4C), and did not reduce the mRNA expression of *MCL1* (Figure 4D). Bax expression did not change after proscillaridin A treatment (Figure 4C). As the activation of caspase 8 did not change in SW620 cells, this reduction in Mcl1 expression suggests that proscillaridin A enhanced TRAIL-induced apoptosis via the mitochondrial pathway downstream of caspase 8 activation in SW620 cells. 

### 2.5. The TRAIL-Sensitizing Effect of Proscillaridin A Is Partly Mediated via O-Glycosylation

The expression of *GALNT14* or *C1GALT1C1*, which are involved in the generation of *O*-glycans, promotes TRAIL-induced caspase 8 activation and cell death [13,14,15]. We found that proscillaridin A significantly enhanced the expression levels of these genes and *C1GALT1*, which is critical for the synthesis of type 1 and 2 *O*-glycans, in HT29 cells (Figure 5A). To examine whether the upregulation of these enzymes contributes to the TRAIL-sensitizing effect of proscillaridin A, we treated HT29 cells with benzyl-α-GalNAc, which inhibits *O*-glycan elongation at the initial GalNAc residue [23,24]. Treatment with benzyl-α-GalNAc slightly but significantly suppressed proscillaridin A and TRAIL-induced cell death at the early and middle time points (3–6 h after TRAIL stimulation), but not at the late time points (6–7 h after TRAIL stimulation) (Figure 5B; Appendix A). In addition, proscillaridin A-mediated increase in TRAIL-induced caspase activation was significantly suppressed by benzyl-α-GalNAc treatment (Figure 5C). These results suggest that proscillaridin A sensitizes HT29 cells to TRAIL-induced cell death partly via *O*-glycosylation.

## 3. Discussion

In this study, during the screening of thousands of chemical compounds for the purpose of identifying TRAIL sensitizers, we found several compounds that enhanced TRAIL-induced cell death in colon cancer cells without toxicity as a single agent. A previous study reported that celastrol, a plant-derived triterpene, enhances TRAIL-induced apoptosis by inhibiting TGF-β-activated kinase 1 (TAK1)-mediated nuclear factor-κB activation [25]. As cIAP-mediated ubiquitination of TRAIL-DISC components is required for TAK1 activation, the mechanism of the TRAIL-sensitizing effect of celastrol is similar to that of SMAC mimectis. Adapalene is a synthetic retinoid that is used to treat acne. Retinoic acid enhances TRAIL-induced apoptosis by upregulating DR4 expression levels in lung cancer and head and neck squamous cancer cell lines [26]. However, adapalene did not upregulate DR4 expression in HT29 cells. Similarly, although mitoxantrone enhances TRAIL-induced apoptosis by upregulating DR4 expression levels in glioblastoma and prostate cancer cells [27,28,29], it did not affect DR4 expression in HT29 cells. These results indicate that these agents enhance TRAIL-induced apoptosis via multiple mechanisms in a cell type-specific manner. BIO-acetoxime is a selective inhibitor of GSK-3α/β, and its inhibition has been reported to enhance TRAIL-induced cell death [30,31,32,33,34]. CP-100356 is a P-glycoprotein inhibitor that was previously unknown as a TRAIL sensitizer. 

Among the hit compounds, proscillaridin A markedly enhanced TRAIL-induced cell death. Proscillaridin A is a plant-derived cardiac glycoside that inhibits Na^+^/K^+^-ATPase activity [35]. In addition to its therapeutic effect in cardiovascular diseases, proscillaridin A and other cardiac glycosides have been found to show antitumor effects in various types of cancer [36,37]. Currently, a dozen clinical trials of cardiac glycosides in patients with cancer are underway. Previous studies have shown that other cardiac glycosides, such as digoxin, bufalin, and ouabain, enhance TRAIL-induced cell death in various types of cancer cells, including lung, bladder, liver, and breast cancers, and glioblastoma [38,39,40,41,42,43,44]. The minimum effective concentration of these cardiac glycosides as TRAIL sensitizers differs greatly among each compound (0.5 nM to over 100 nM). Our results showed that proscillaridin A significantly enhanced TRAIL-induced cell death, even at around 10 nM, suggesting that the TRAIL-sensitizing effect of proscillaridin A was either similar to or greater than that of other cardiac glycosides. In addition, proscillaridin A enhanced TRAIL-induced cell death even in the presence of the SMAC mimetic and vice versa, indicating that these two agents have distinct mechanisms to sensitize cells to TRAIL-induced cell death. Indeed, 1 h preincubation with proscillaridin A did not sensitize HT29 cells to TRAIL-induced cell death, whereas it was sufficient for SMAC mimetics. These results suggest that in contrast to SMAC mimetics, which immediately degrade cIAPs via direct interaction and sensitize cells to cell death, proscillaridin A enhances TRAIL-induced cell death via multiple events including, for instance, gene expression.

The mode of action of cardiac glycosides is well defined: inhibition of Na^+^/K^+^-ATPase increases cellular sodium ions, leading to an increase in cellular calcium ions and cardiac contractile force [35]. An increase in cellular calcium ions not only induces actin-myosin interaction, but also affects many other cellular events, such as protein kinase signaling and gene expression [45]. Indeed, an increase in cellular calcium ions enhances TRAIL-induced cell death through upregulation of TRAIL receptors and mitochondrial outer membrane permeability [46,47,48]. In addition, the binding of cardiac glycosides to Na^+^/K^+^-ATPase activates various calcium-independent signaling pathways [49]. For instance, activation of the Src-MAPK pathway by cardiac glycosides leads to the generation of reactive oxygen species and thereby sensitizes cells to cell death [50,51]. Therefore, cardiac glycosides dramatically alter cellular states through multiple mechanisms. A previous study showed that proscillaridin A dramatically alters the transcriptional profile [52,53]. Here, we observed the transcriptional upregulation of TRAIL receptors and enzymes involved in *O*-glycosylation, such as *GALNT14*, *C1GALT1*, and *C1GALT1C1*. The expression of these molecules is regulated by multiple mechanisms, including transcription factors, epigenetic modifications, and miRNAs [54,55]. Therefore, proscillaridin A may upregulate TRAIL receptor expression and *O*-glycosylation by affecting multiple pathways, including histone acetylation and MYC expression. 

Proscillaridin A treatment reduced the protein expression levels of cFLIP_L_ in SW480 cells and Mcl1 in SW620 cells, respectively. However, this reduction in protein expression was not accompanied by a decrease in mRNA expression. cFLIP_L_ and Mcl1 are highly labile proteins with a half-life of a few hours and are mainly degraded by the ubiquitin-proteasome pathway [21,56]. Proscillaridin A and other cardiac glycosides were previously reported to cause the degradation of Mcl1 via the ubiquitin-proteasome pathway [57]. Therefore, the reduction in cFLIP_L_ and Mcl1 levels by proscillaridin A may be due to ubiquitin-mediated proteasomal degradation. 

*O*-glycosylation has been implicated in the regulation of TRAIL-induced caspase 8 activation and cell death [12]. Previous studies have reported that *DR5* is *O*-glycosylated at multiple serine/threonine residues and that the expression of *GALNT14* or *C1GALT1C1* promotes ligand-dependent *DR5* oligomerization [13]. However, manipulation of the expression of these enzymes affects the expression level and structure of *O*-glycans on not only *DR5* but also other *O*-glycosylated proteins. Therefore, it remains unclear whether *O*-glycans on *DR5* regulate ligand-dependent *DR5* oligomerization. Future studies are required to elucidate the detailed molecular mechanisms underlying *O*-glycan-mediated regulation of TRAIL-induced cell death.

As shown in this and the abovementioned previous studies, proscillaridin A affects diverse cellular processes, such as transcription and protein degradation. As cancer cells acquire resistance to TRAIL therapeutics via multiple mechanisms, proscillaridin A could have better efficacy in sensitizing cells to TRAIL therapeutics than agents targeting a single mechanism. Thus, co-treatment with proscillaridin A and TRAIL, in combination with SMAC mimetics in some cases, may be used as a novel therapeutic strategy for cancer treatment.

## 4. Materials and Methods

### 4.1. Cell Lines

Human colon cancer cell lines (HT29, SW480, SW620, LS174T, Lovo, HCA7, and Caco2) were cultured in Dulbecco’s modified Eagle’s medium (DMEM) supplemented with 10% fetal bovine serum (FBS), 100 units/mL of penicillin, and 100 µg/mL of streptomycin. 

### 4.2. Reagents

The following compounds were used in this study: acebutolol (23393, Cayman Chemical, Ann Arbor, MI, USA), adapalene (A2549, Tokyo Chemical Industries, Tokyo, Japan), BIO-acetoxime (16329, Cayman Chemical), caspofugin acetate (15923, Cayman Chemical), celastrol (70950, Cayman Chemical), cepharanthine (19648, Cayman Chemical), colistin sulfate (AG-CN2-0065-M100, Adipogen, San Diego, CA, USA), CP-100356 monohydrochloride (PZ0171, Sigma-Aldrich, Burlington, MA, USA), dabigatran etexilate (17131, Cayman Chemical), desloratadine (16931, Cayman Chemical), diethylstilbestrol (10006876, Cayman Chemical), fluoxetine hydrochloride (CDX-F0141-M010, Adipogen), fluspirilene (19530, Cayman Chemical), methiothepin maleate (23138, Cayman Chemical), mitoxantrone hydrochloride (14842, Cayman Chemical), N-allylthiourea (A0220, Tokyo Chemical Industries), plerixafor hydrochloride hydrate (10011332, Cayman Chemical), proscillaridin A (0852, EXTRASYNTHESE, Genay, France), ranolazine hydrochloride (15604, Cayman Chemical), ruthenium red (14339, Cayman Chemical), and zoxazolamine (320-86751, Wako, Osaka, Japan). Benzyl-α-GalNAc was obtained from Sigma-Aldrich.

### 4.3. Drug Screening

The drug library was composed of the Spectrum Collection (2320 compounds), Phizer Drug (90 compounds), Tocriscreen Stem Cell Toolbox (80 compounds), LoPac 1280 (1280 compounds), and FDA-approved drug library (1134 compounds) provided by the Center for Supporting Drug Discovery and Life Science Research at Osaka University Graduate School of Pharmaceutical Science. The compounds were dispensed in 384-well plates (250 nl/well) and stored at −30 °C until use. One day before stimulation, HT29 cells were inoculated at 4.5 × 10^3^ cells (30 µL)/well in 384-well plates containing the compounds by using a Multidrop Combi Reagent Dispenser (Thermo Fisher Scientific, Waltham, MA, USA). Ten µL of 2.5 µM birinapant (TL-32711) (Active BioChem, Hong Kong, China) in 2.5% DMSO solution was added to wells by Certus Nano Liquid Flex 5 (Gyger Fluidies, Gwatt, Switzerland). After preincubation for 30 min, 10 µL of 7.5 ng/mL TRAIL (BML-SE721, Enzo Life Sciences, Madison, NY, USA) in culture medium was added to wells with Certus Nano Liquid Flex 5 and cells were cultured for 24 h. Final concentrations of compounds, birinapant, DMSO, and TRAIL were 5 µM, 0.5 µM, 1.0%, and 1.5 ng/mL, respectively. The culture medium was completely discarded, and 30 µL of 2-fold diluted CellTiter-Glo Luminescent Cell Viability Assay Solution (G7570, Promega, Madison, WI, USA) was added to each well with a Multidrop Combi Reagent Dispenser. After 10 min of incubation at room temperature, chemiluminescence was measured by using a GloMax Discover Multimode Microplate Reader (Promega). In each screening plate, wells without any treatment (0% cell death) and with only 50 ng/mL TRAIL (100% cell death) were prepared and used to determine cell viability in wells of interest. 

### 4.4. Cell Death Assay

One day before stimulation, the cells were inoculated in a 96-well plate. Cells were treated with the indicated concentrations of TRAIL in FluoroBrite DMEM Media supplemented with 10% FBS, 4 mM glutamate, 1 mM pyruvate, 100 units/mL penicillin, and 100 µg/mL streptomycin, 0.625 µg/mL Hoechst 33342 (Sigma-Aldrich), and 5 µg/mL PI (Sigma-Aldrich). Where indicated, cells were treated with compounds for 24 h and/or 0.5 µM BV6 (APExBIO, Houston, TX, USA) for 1 h before and during TRAIL stimulation. Cells were imaged at intervals of 15 min by using Operetta CLS (Perkin Elmer, Waltham, MA, USA). Cell death was determined as the percentage of PI positivity, which is the ratio of the number of PI-positive cells to the total cell number in a well, determined by staining with Hoechst 33342. For the ATP assay, cells were stimulated in DMEM supplemented with 10% FBS, 100 units/mL penicillin, and 100 µg/mL streptomycin. The intracellular ATP was determined by using CellTiter-Glo luminescent cell viability assays, according to the manufacturer’s instructions. For annexin V/PI staining, cells were stained with 1.8 µg/mL FITC-annexin V (BioLegend, San Diego, CA, USA) and 5 µg/mL PI for 10 min in annexin V staining buffer (10 mM HEPES, 140 mM NaCl, 2.5 mM CaCl_2_) and analyzed by using FACS Canto II or LSR Fortessa X-20 (BD Biosciences, Franklin Lakes, NJ, USA). 

### 4.5. Flow Cytometry

Cells were detached from tissue culture dish by incubation with 2 mM ethylenediaminetetraacetic acid (EDTA)/phosphate-buffered saline (PBS) and stained with first antibodies in 1% bovine serum albumin (BSA)/PBS for 30 min at 4 °C. As a negative control, cells were incubated with mouse IgG (MO75-3, MBL, Tokyo, Japan). After washing with 1% BSA/PBS, cells were incubated with secondary antibodies in 1% BSA/PBS for 30 min at 4 °C. After cells were washed with 1% BSA/PBS, stained cells were analyzed by using BD LSRFortessa X-20 (BD Biosciences) and FlowJ software (BD Biosciences). The primary antibodies used for flow cytometry were anti-DR4 antibody (Sigma-Aldrich, SAB4700541, DR-4-02, 4 µg/mL) and anti-*DR5* antibody (Thermo Fisher Scientific, 14-9909-82, DJR2-2 (2-6), 2 µg/mL). The secondary antibody used for flow cytometry was Alexa Fluor 488-conjugated anti-mouse IgG (715-546-151, 1/250, Jackson Laboratories, Bar Harbor, ME, USA). 

### 4.6. Western Blotting

Cells were lysed by using the radioimmunoprecipitation assay buffer (50 mM Tris-HCl, 150 mM NaCl, 1% NP40, 0.5% sodium deoxycholate, 0.1% sodium dodecyl sulfate [SDS]). Lysates were subjected to SDS-polyacrylamide gel electrophoresis. Proteins were transferred onto polyvinylidene fluoride membrane (Millipore, Burlington, MA, USA). The membrane was blocked with 5% skim milk (Nacalai, Kyoto, Japan) in TBS supplemented with 0.05% Tween (TBST) at room temperature for 1 h and then incubated with primary antibodies at 4 °C overnight. After washing with TBST, the membranes were incubated with secondary antibodies. The membrane was washed with TBST and developed by using SuperSignal West Dura (Thermo Fisher Scientific). The signal was visualized by using Amersham Imager 600 (Cytiva, Marlborough, MA, USA). The following first antibodies were used: anti-Bax (Santa Cruz Biotechnology, Santa Cruz, CA, USA, sc-7480, 1/200), anti-Bcl-xL (BD Biosciences, 610747, 1/1000), anti-caspase 3 (Cell Signaling Technology (CST), Danvers, MA, USA, 9662, 1/1000), anti-caspase 8 (CST, 9746, 1/1000), anti-cFLIP (Enzo Life Sciences, 7F10, 1/1000), anti-cIAP1 (Enzo Life Sciences, 1E1-1-10, 1/1000), anti-cleaved PARP (CST, 9541, 1/1000), anti-CYLD (Santa Cruz Biotechnology, sc-74435, 1/1000), anti-FADD (Millipore, 1F7, 1/1000), anti-glyceraldehyde-3-phosphate dehydrogenase (anti-*GAPDH*; Santa Cruz Biotechnology, sc-32233, 1/100,000), anti-IKKα/β (Santa Cruz Biotechnology, sc-7607, 1/200), anti-Mcl1 (ProteinTech, Rosemont, IL, USA, 16225-1-AP, 1/1000), anti-RIPK1 (BD Biosciences, 610459, 1/1000), anti-TRAF2 (Santa Cruz Biotechnology, sc-876, 1/500), and anti-XIAP (MBL, M044-3, 1/1000) antibodies. Horseradish peroxidase (HRP)-conjugated donkey anti-rabbit IgG (Cytiva, NA934, 1:5000), HRP-conjugated sheep anti-mouse IgG (Cytiva, NA931, 1:5000), and HRP-conjugated goat anti-rat IgG (Cytiva, NA935, 1:5000) secondary antibodies were used.

### 4.7. Quantitative Reverse Transcription (qRT)-Polymerase Chain Reaction (PCR)

Total RNA was extracted from cells by using NucleoSpin RNA (Macherey Nagel, Düren, Germany) and subjected to RT by using ReverTra Ace (TOYOBO, Osaka, Japan). Real-time PCR was performed with Fast SYBR Green master mix (Thermo Fisher Scientific) by using a QuantStudio 3 real-time PCR system (Thermo Fisher Scientific). The data were analyzed by using the comparative Ct method. *GAPDH* was used as an internal control to normalize the expression levels of the target genes. The primer sequences used in this study are listed in Table 1. 

### 4.8. Statistical Analysis

Statistical analysis was performed by using an unpaired *t*-test with Welch’s correction and one-way or two-way analysis of variance by using Prism 9 software. *p* < 0.05 was considered to be statistically significant. 

## Figures and Tables

**Figure 1 ijms-23-06973-f001:**
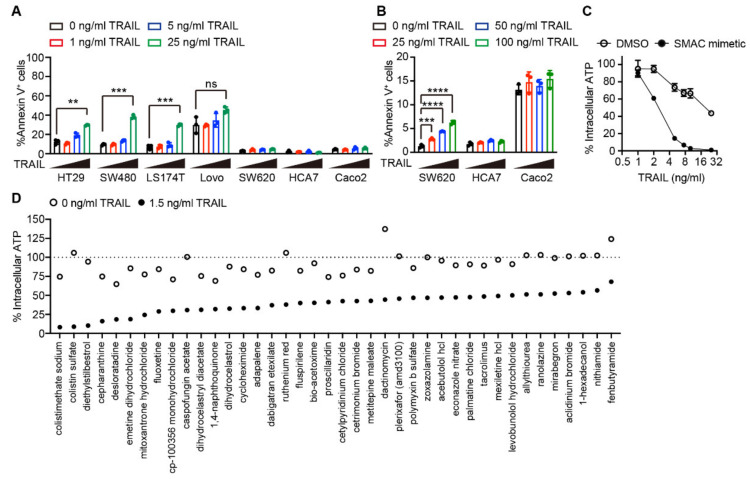
Screening a chemical library to identify agents to sensitize cells to tumor necrosis factor-related apoptosis-inducing ligand (TRAIL)-induced cell death. (**A**,**B**) Various colon cancer cells were treated with TRAIL at the indicated concentration for 16 h. The percentage of annexin V^+^ cells was determined via flow cytometry. (**C**) HT29 cells were pretreated with 0.5 µM second mitochondria-derived activator of caspase (SMAC) mimetic (birinapant) for 30 min and then treated with TRAIL at the indicated concentration for 24 h. The intracellular ATP level was determined via CellTiter-Glo luminescent cell viability assay. (**D**) HT29 cells were treated with either chemical compounds alone (white circle) or the compounds, birinapant, and TRAIL (black circle) as described in Materials and Methods. The intracellular ATP level was determined as in (**C**). Results are represented as the mean ± standard deviation (SD) of triplicate samples (**A**–**C**). ns, not significant. ** *p* < 0.01; *** *p* < 0.001, **** *p* < 0.0001 (unpaired *t*-test with Welch’s correction in (**A**) and one-way analysis of variance (ANOVA) in (**B**)).

**Figure 2 ijms-23-06973-f002:**
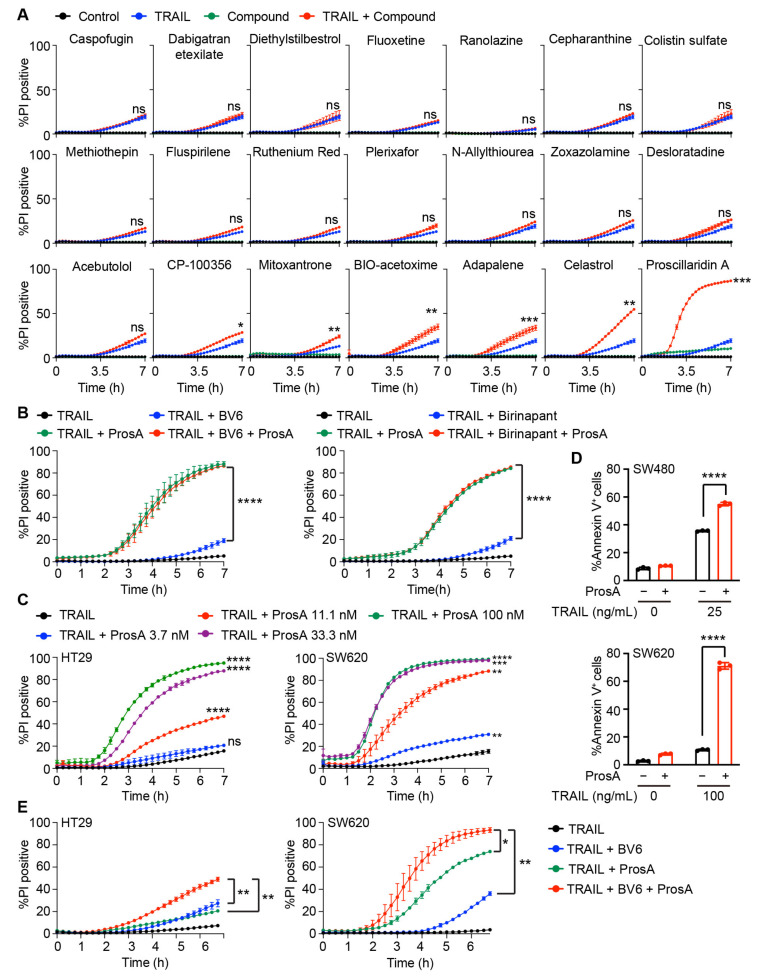
Proscillaridin A enhances TRAIL-induced cell death. (**A**) HT29 cells were pretreated with 1 µM of various compounds for 24 h and then treated with 5 ng/mL TRAIL. Statistical analysis was performed by using two-way ANOVA (TRAIL alone vs. TRAIL + Compound). (**B**) HT29 cells were pretreated with 1 µM proscillaridin A (ProsA) for 24 h. Subsequently, the cells were treated with 0.5 µM SMAC mimetic (BV6 or birinapant) for 1 h and further treated with 5 ng/mL TRAIL for the indicated time. Statistical analysis was performed by using two-way ANOVA (TRAIL + SMAC mimetic vs. TRAIL + SMAC mimetic + ProsA). (**C**) HT29 and SW620 cells were pretreated with the indicated concentration of proscillaridin A for 24 h and then treated with TRAIL at 10 and 33 ng/mL for HT29 and SW620 cells, respectively. Statistical analysis was performed by using two-way ANOVA (vs. TRAIL alone shown in black). (**D**) SW480 and SW620 cells were pretreated with 11.1 nM proscillaridin A for 24 h and then treated with TRAIL for 16 h. The percentage of annexin V^+^ cells was determined via flow cytometry. Statistical analysis was performed via an unpaired *t*-test with Welch’s correction. (**E**) HT29 and SW620 cells were pretreated with 11.1 nM proscillaridin A for 24 h and then 0.5 µM BV6 for 1 h. After pretreatment, the cells were treated with TRAIL at 1 and 11 ng/mL for HT29 and SW620 cells, respectively. Statistical analysis was performed by using two-way ANOVA (TRAIL + BV6 or ProsA vs. TRAIL + BV6 + ProsA). The percentage of PI^+^ cells was determined by using Operetta CLS (**A**–**C**,**E**). Results are represented as the mean ± SD of triplicate (right panel in (**B)**, (**D**), HT29 in (**C**)) or duplicate ((**A**), left panel in (**B**), SW620 in (**C**), (**E**)) samples. Results are representatives of two independent experiments (**A**–**E**). ns: not significant. * *p* < 0.05; ** *p* < 0.01; *** *p* < 0.001; **** *p* < 0.0001.

**Figure 3 ijms-23-06973-f003:**
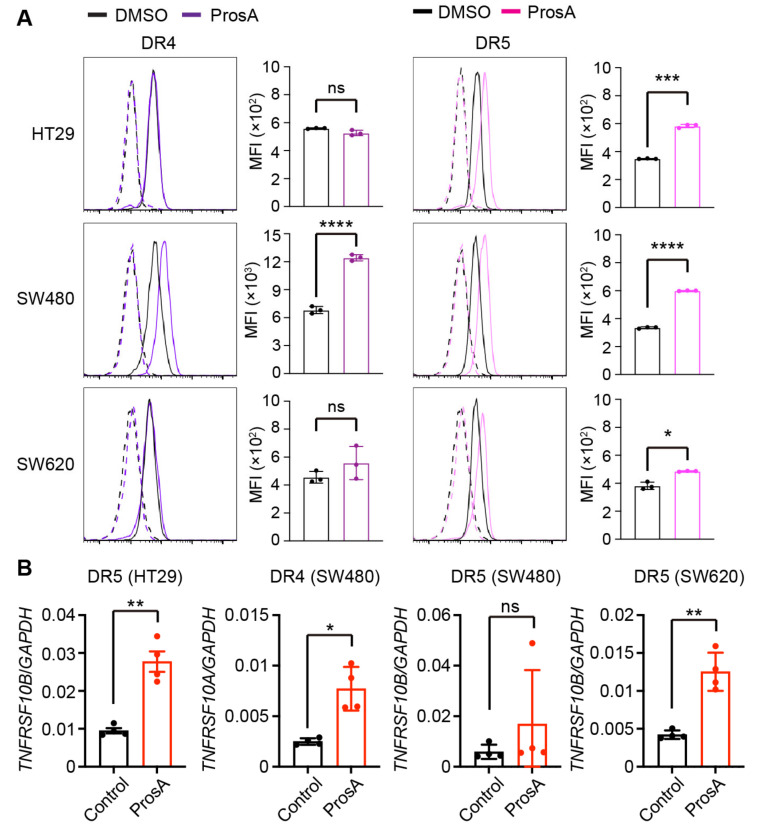
Proscillaridin A upregulates the expression of TRAIL receptors in a cell type-specific manner. (**A**) HT29, SW480, and SW620 cells were treated with 11.1 nM proscillaridin A (ProsA) for 24 h. Cell surface expression levels of DR4 and *DR5* were determined via flow cytometry. Dotted lines indicate the cells stained without primary antibodies. Representative histograms from three independent experiments are shown. Bar-dot graphs indicate the mean fluorescence intensity (MFI). Results are represented as the mean ± SD of triplicate. (**B**) Relative gene expression levels of TNF receptor superfamily member 10a (*TNFRSF10A*; DR4) and *TNFRSF10B* (*DR5*) was determined via quantitative reverse transcription-polymerase chain reaction (qRT-PCR). The expression levels of the target gene were normalized to that of glyceraldehyde-3-phosphate dehydrogenase (*GAPDH*). Results pooled from two independent experiments are shown (n = 4). Results are represented as the mean ± SD. * *p* < 0.05; ** *p* < 0.01, *** *p* < 0.001; **** *p* < 0.0001 (unpaired *t*-test with Welch’s correction). ns, not significant.

**Figure 4 ijms-23-06973-f004:**
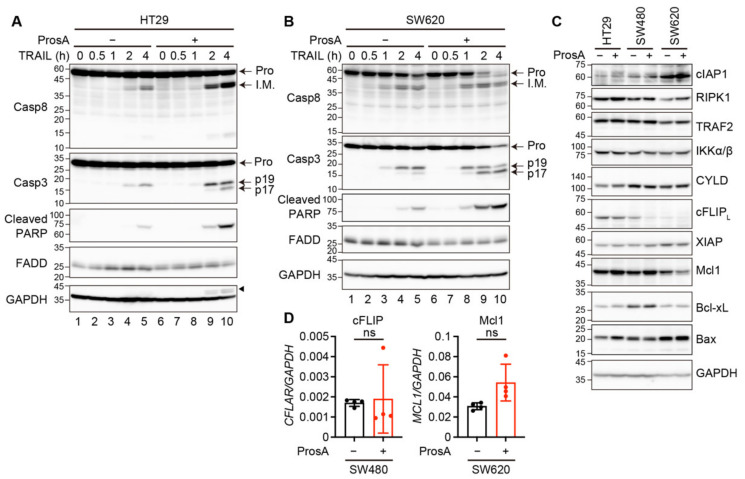
Proscillaridin A enhances TRAIL-induced caspase activation. (**A**,**B**) HT29 and SW620 cells were pretreated with 11.1 nM proscillaridin A (ProsA) for 24 h and then treated with TRAIL at 5 and 100 ng/mL for HT29 and SW620 cells, respectively. An arrowhead indicates the residual signal from the previous blot (Casp8). (**C**) The cells were treated with 11.1 nM ProsA for 24 h. Whole cell extracts were subjected to Western blotting. Pro, proform; I.M., intermediate form. Blots are representatives of two independent experiments (**A**–**C**). (**D**) Relative gene expression levels of cellular FADD-like IL-1β converting enzyme-like inhibitor protein (*CFLAR*/cFLIP) and *MCL1* were determined via qRT-PCR. The expression levels of target genes were normalized by that of *GAPDH*. Results pooled from two independent experiments are shown (n = 4). Results are represented as the mean ± SD. ns, not significant.

**Figure 5 ijms-23-06973-f005:**
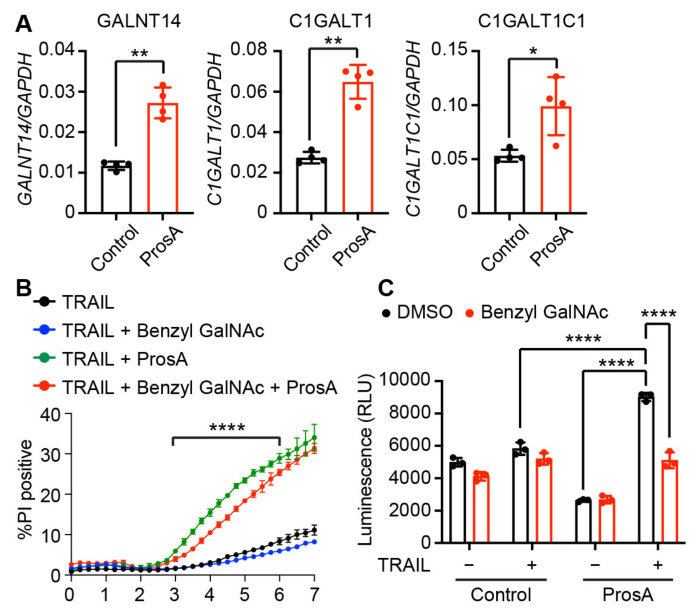
Proscillaridin A enhances TRAIL-induced cell death partly via *O*-glycosylation. (**A**) Relative gene expression in HT29 cells treated with 11.1 nM proscillaridin A (ProsA) for 24 h was determined via qRT-PCR. The expression levels of target genes were normalized to that of *GAPDH*. Results pooled from two independent experiments are shown (n = 4, mean ± SD). (**B**) HT29 cells were pretreated with 11.1 nM ProsA and/or 2 mM benzyl-α-GalNAc for 24 h and then treated with 20 ng/mL TRAIL. The percentage of PI^+^ cells was determined by using Operetta CLS. Statistical analysis was performed for TRAIL + ProsA and TRAIL + Benzyl GalNAc + ProsA (3–6 h after TRAIL stimulation). (**C**) HT29 cells were pretreated with 33.3 nM proscillaridin A and/or 2 mM benzyl-α-GalNAc for 24 h and then treated with 25 ng/mL TRAIL for 1.5 h. Caspase 8 activity was determined via a Caspase-Glo8 assay. Results are represented as the mean ± SD of triplicate (**B**,**C**). Results are representative of two independent experiments (**B**,**C**). * *p* < 0.05; ** *p* < 0.01; **** *p* < 0.0001 (unpaired *t*-test with Welch’s correction in (**A**) and two-way ANOVA in (**B**,**C**)).

**Table 1 ijms-23-06973-t001:** Sequence of primers used in this study.

Genes	Forward	Reverse
*C1GALT1*	AAAGGCCAAACACGTCAAAG	GCCTTCTTTGGTTTTCAGTCC
*C1GALT1C1*	TTGAAGGGTGTGATGCTTGG	ATGCTCATGGTGGTGCATTC
*CFLAR*	GACAGAGCTTCTTCGAGACAC	GCTCGGGCATACAGGCAAAT
*DR5*	GCCCCACAACAAAAGAGGTC	AGGTCATTCCAGTGAGTGCTA
*GALNT14*	TGCCCAAGGTGAAATGCTTG	TCGAGGAAAGTCAGAGTGGTG
*GAPDH*	GAAATCCCATCACCATCTTCCAGG	GAGCCCCAGCCTTCTCCATG
*MCL1*	TGCTTCGGAAACTGGACATCA	TAGCCACAAAGGCACAAAAG

## Data Availability

Data is contained within the article or Appendix A.

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
