# Peer review of "Proscillaridin A Sensitizes Human Colon Cancer Cells to TRAIL-Induced Cell Death"

_ijms, 2022, doi:10.3390/ijms23136973_

Round 1

Reviewer 1 Report

The manuscript entitled " Proscillaridin A Sensitizes Human Colon Cancer Cells to  TRAIL-Induced Cell Death " was thoroughly reviewed. This work is well executed and well written, and the results were impressive. 

This article is well planned; however, there are some questions to be clarified:

1.      Figure 1D:

Birinapan and TRAIL + birinapan should be mentioned in the legend to the figure, otherwise it is confusing why TRAIL showed the highest activity with proscillardin in the second round of screening, while in the first screen, the activity of proscillardin is in 20th place in combination with TRAIL.

Figure 1 legend:  ns: not significant –no indication on the Figure 1.

2.      Why the authors used Birinapant as a Smac mimetic in compound selection experiments and switched to BV6 to study the synergistic effect of proscillaridin A on TRAIL.

3.      Line 144 – “Proscillaridin A significantly enhanced TRAIL-induced cell death in the presence of a SMAC mimetic”. According to the data shown in Fig. 2B, proscillaridin A significantly enhanced TRAIL-induced cell death in both the absence and presence of the SMAC mimetic.

4.      The conditions for the selection of drugs vary greatly in Figs. 1 and Fig. 2. In Fig. 1, the concentrations of compounds and TRAIL were 0.5 μM and 1.5 ng/mL, respectively, and the SMAC mimetic was used, while in Fig. 2 they were 1 μM. and 5 ng/ml for compounds and TRAIL, respectively, without the SMAC mimetic. The time of incubation with TRAIL also differs (24 and 7 h, respectively). Perhaps that is why the compounds that were more effective in Fig. 1 do not show synergy for TRAIL in Fig. 2. There is no explanation for this in the article.

5.      The authors incubated the cells for 24 hours with proscillaridine A before adding TRAIL. Did the authors attempt to pre-incubate the cells for a shorter time with proscillaridin A, or whether they investigated the combined development of proscillaridin A and TRAIL without pre-incubation?

Reviewer 2 Report

The paper of Manami Semba et al. deals with the hot topic of TRAIL-related combined cancer therapies. The Authors demonstrate the synergy of attenuating TRAIL and Proscillaridin A effects on the viability of cancer cells. The paper is well written and conceptually sound. There are, however, two points to be addressed by the Authors before the final submissiom:

- ATP assays hardly estimate the %age of living/dead cells. This point should be corrected throughout the text/axis descriptions in the figures;

- are there any links between the activity of NA/K ion channel/pump and apoptosis-related intracellular signaling?
